# Involvement of Actin and Actin-Binding Proteins in Carcinogenesis

**DOI:** 10.3390/cells9102245

**Published:** 2020-10-06

**Authors:** Magdalena Izdebska, Wioletta Zielińska, Marta Hałas-Wiśniewska, Alina Grzanka

**Affiliations:** Department of Histology and Embryology, Faculty of Medicine, Collegium Medicum in Bydgoszcz, Nicolaus Copernicus University in Toruń, 85-092 Bydgoszcz, Poland; w.zielinska@cm.umk.pl (W.Z.); mhalas@cm.umk.pl (M.H.-W.); agrzanka@cm.umk.pl (A.G.)

**Keywords:** actin, ABPs, carcinogenesis, EMT, metastasis, angiogenesis, vasculogenic mimicry

## Abstract

The actin cytoskeleton plays a crucial role in many cellular processes while its reorganization is important in maintaining cell homeostasis. However, in the case of cancer cells, actin and ABPs (actin-binding proteins) are involved in all stages of carcinogenesis. Literature has reported that ABPs such as SATB1 (special AT-rich binding protein 1), WASP (Wiskott-Aldrich syndrome protein), nesprin, and villin take part in the initial step of carcinogenesis by regulating oncogene expression. Additionally, changes in actin localization promote cell proliferation by inhibiting apoptosis (SATB1). In turn, migration and invasion of cancer cells are based on the formation of actin-rich protrusions (Arp2/3 complex, filamin A, fascin, α-actinin, and cofilin). Importantly, more and more scientists suggest that microfilaments together with the associated proteins mediate tumor vascularization. Hence, the presented article aims to summarize literature reports in the context of the potential role of actin and ABPs in all steps of carcinogenesis.

## 1. Introduction

The cytoskeleton participates in many physiological processes. It is responsible for cell movement, division, differentiation, senescence, and death. One of the main components of the structure are actin filaments, without which cells are unable to move, divide, and die through apoptosis [1]. Moreover, nuclear actin participates in the regulation of gene expression as a component of chromatin remodeling complexes. Hence, it is one of the pivotal factors in the maintenance of cell homeostasis [2].

The inseparable elements in the context of the actin cytoskeleton are ABPs (actin-binding proteins). They influence the dynamics of actin filaments by promoting their polymerization and degradation [3]. It may occur in many ways. For example, through creating new polymerization sites (Arp2/3 complex (actin-related protein 2/3 complex)), branching and crosslinking ((Arp 2/3 complex, WASP (Wiskott-Aldrich syndrome protein), fascin, α-actinin, TAGLN (transgelin)), stabilization of the actin structure (tropomyosin), blocking of the free ends (GLS (gelsolin), villin, formin), promoting the delivery of globular monomers (profilin), or severing and depolymerization (CFL (cofilin), GLS) [4,5]. Therefore, the binding of ABPs by actin filaments has to be strictly controlled by the action of intracellular mechanisms and signaling pathways related, to e.g., local changes in the concentration of calcium ions. 

Reorganization of the actin cytoskeleton with the participation of ABPs is inherent in invasion and metastasis. Moreover, the proteins create a link between the cytomembrane and nucleus, influencing the nuclear actin pool, and thus, gene expression. It may also affect the response of cells to the action of cytostatics [6]. Furthermore, the mechanism of movement associated with the reorganization of the cytoskeleton is universal for both normal and cancer cells, regardless of the type of tumor. For these reasons, manipulating the level of ABPs seems to be an attractive approach in the context of inhibiting tumor progression.

In this review, we focus on the participation of actin and ABPs in carcinogenesis and their influence on cancer progression and tumor vascularization. Moreover, we describe potential therapeutic targets associated with the inhibition of these processes through actin cytoskeleton manipulation.

## 2. Cytoskeleton, Its Structure, Function, and Regulation

Actin filaments are commonly associated with muscle cells. However, they are also present in nonmuscle ones in which they concentrate mostly in the cortex. Moreover, adherent cells can form stress fibers anchored to the plasma membrane in response to stress conditions or during cell movement. F-actin (fibrillar actin) is a long-chain polar polymer formed as a result of the polymerization of G-actin (globular actin). However, during the polymerization, necessary elements are not only monomers but also ATP and ABPs [7]. The first step of the filament formation process is nucleation, during which three G-actin monomers form a stable nucleation center with the participation of the Arp2/3/WASP complex. Two of the seven subunits of the protein—Arp2 and Arp3—are characterized by a great homology with G-actin. Early studies on the topic suggested that Arp2-Arp3 stable dimer may mimic the unstable actin–actin complex necessary for the initiation of nucleation [8]. However, further research showed that the process is way more complex and requires additional regulators [9]. Arp2/3 is also a pivotal factor in the creation of branched actin filaments. The protein can attach to the already formed microfilaments under a constant angle of approximately 70° [10]. It initiates the assembly of a new filament on the side of the previous one. This process is strictly regulated by the NPFs (nucleation promoting factors). One of the NPFs is the proteins belonging to the Wiskott–Aldrich syndrome family (WASP, N-WASP, WAVE, and WASH). They contain conservative VCA (verprolin, cofilin, and acidic) domain able to activate the Arp2/3 complex through the introduction of conformational changes. The proteins also deliver the first actin monomer, which builds the daughter filament [11]. Another factor involved in the creation of a branched actin network is cortactin as it can directly activate Arp2/3 but also stabilize new filament branch points [12]. As demonstrated both in the case of theoretical models and experimental studies, the intensity of branching is closely related to the speed of directional migration [13,14]. This is due to the fact that the branched structure of actin participates in the process of creating invasive structures such as lamellipodia and invadopodia.

The next phase, during which G-actin attaches to the barbed end of the actin filament, is elongation. One of the consequences is a local reduction in G-actin levels. When the G-actin availability decreases, the barbed end still grows, but at the same time, the pointed end drops in length. The phase will continue until the elongation rate is greater than the loss of ADP-actin from the pointed end [15]. Another protein responsible for the regulation of actin polymerization is profilin, which binds to the barbed end of actin, catalyzes the exchange of ADP/ATP, and thereby increases the polymerization of actin filaments [16]. The protein binds to G-actin, driving the filament assembly but only at the barbed end. Profilin-G-actin complexes interact with formins, which facilitates their recruitment to the fast-growing end. It ensures a dynamic balance between the pool of profilin-G-actin and F-actin. Another G-actin binding protein is Tβ4 (thymosin-β4), involved in the inhibition of F-actin polymerization. The competition between Tβ4 and profilin constantly regulates the pace and effectiveness of this process [17].

The group of nucleation regulators also includes formin and proteins with tandem formin homology domains FH1 and FH2. The first one induces the elongation of unbranched actin filaments by binding to the actin nuclei or barbed ends. In turn, the latter affects nucleation by promoting the association of dimers or trimers [18,19]. Formins are also involved in actin polymerization. The participation of the protein in actin filament assembly is particularly interesting, considering its sensitivity to tensile forces. It suggests that they are associated with the cellular response to mechanical stress. Formins in an active form occurring as a dimer, which creates a sleeve surrounding the actin subunits. The active complex can compete with capping proteins and substantially remove them. Simultaneously, it prevents recapping allowing the filament to grow or form cross-links. Moreover, through the interaction with profilin-bounded actin monomers, it facilitates the filament elongation [20]. Proteins from the formin family are of great importance in proper cell division. mDia2 is involved in the formation of the contractile ring [21]. In turn, mDia3 participates in the correct segregation of chromosomes in the course of mitosis [22]. However, mDia1 and mDia2 are also the regulators of cell mobility. Both of the proteins are strictly connected with filopodia formation driven by the RhoGTPases [23]. Another member of the formins family is FMNL1 (formin-like 1), engaged in the maintenance of proper actin dynamic ensuring the stability of the Golgi apparatus [24]. Interestingly, formins may also be involved in the changes in cell morphology characteristic for invasive cancer cells. FHOD1 (Formin Homology 2 Domain Containing 1) upregulation results in F-actin organization characteristic for mesenchymal cells together with spindle-like shape [25].

Except for elongation, ABPs can also cap barbed or pointed ends, cut filaments, and cross-link them [26]. The severing and capping proteins that target the actin filaments and regulate their length include CFL, villin, formin, and proteins of the GLS family. In the presence of calcium ions, GLS severs the actin filaments leading to the formation of caps at their barbed ends. This is due to the sensitivity of segments 4–6 to calcium ions, whose binding causes a structural change in the protein. Moreover, it is likely that the capping efficiency increases with the rising concentration of calcium ions [4]. Capping is reversible and can switch into directed polymerization. In turn, CFL action is far more complicated. It participates not only in the cutting of actin filaments but also takes part in the import of actin to the nucleus [27]. Another aspect is the functional relationship of the ADF (actin-depolymerizing factor)/CFL family with environmental factors such as the presence of divalent ions or pH [4]. There are three isoforms of CFL. CFL1 is identified in all mammalian cells, while CFL2 occurs only in muscle ones. In turn, the presence of ADF is characteristic of nerve and epithelial cells. Individual isoforms are characterized by different sensitivity to external factors, which further complicates their role in regulating the dynamics of the actin cytoskeleton. For example, for CFL1 low pH values facilitate faster binding to F-actin, which results in the limitation of depolymerization rate [28]. On the other hand, tropomyosin stabilizes the microfilaments and protects the fibers from severing by CFL [26].

Moreover, there are proteins such as Fln (filamin) and fascin, which organize actin filaments into a cross-linked network of bundles [29]. Proteins from the Fln family can bind both G- and F-actin. However, the most important aspect is its structure, which allows the creation of a flexible connection between actin filaments. It is necessary for the process of the formation of invasive structures like filopodia or lamellipodia [30]. Moreover, proteins of the Fln family are capable of binding many kinds of proteins (e.g., signaling molecules, transmembrane receptors, and cell adhesion molecules). It determines the special role of Fln in the regulation of mechanical stability by transmitting signals between mechanosensitive components and the microfilament network [31]. In turn, fascin, through four specific tandem β-trefoil domains, attaches to actin filaments, taking part in their binding in parallel bundles [32]. Thus, protein is important in the formation of stress fibers, adhesion, and invasion protrusions. Literature reports indicate that a high level of fascin correlates with a deterioration of the survival probability among cancer patients [29]. It emphasizes the potential of the protein as a therapeutic target [29]. Huang et al. showed that the use of a fascin inhibitor (G2) suppresses the collective migration of MDA MB 231 cells [33]. It was also confirmed in a mouse model [29]. Moreover, Wang et al. point out that the blocking of the protein not only inhibits cell migration but also reduces breast cancer tumor size [34]. Due to the optimistic results of preclinical studies, NP-G2-044 fascin inhibitor is currently the only compound influencing the activity of ABPs that has entered the stage of clinical trials (NCT03199586) [35].

It should be emphasized that some of the ABPs may have multiple functions (Figure 1). For example, villin can be associated with nucleation, cross-linking, and capping. Protein action is a consequence of its structure and the concentration of Ca^2+^ [36]. Moreover, PIP2 (phosphatidylinositol 4,5-bisphosphate) causes structural changes in the villin, which also influences its function [36]. Interestingly, Wang et al. suggest the proapoptotic and antiapoptotic impact of villin in small intestine epithelial cells [37]. Another protein, Mena/VASP (mammalian enabled/vasodilator-stimulated phosphoprotein) is important during nucleation and polymerization of actin. Moreover, it is involved in the organization of the actin network in invasive protrusions and cell migration [38].

Additionally to ABPs, the actin reorganization is modulated by small GTP-binding proteins such as Rho, Rac, Ras, and Cdc42 (cell division control protein 42) and kinase-phosphatase pathways [39]. The downstream proteins of activated Rho GTPases are protein kinases and some of the ABPs. As a result, local events leading to the activation of Rho proteins also affect the dynamics of the actin cytoskeleton. An example could be the activation of the Arp2/3 complex by N-WASP and WAVE (WASP family Verprolin-homologous protein). In the case of N-WASP, the Arp2/3-activating VCA domain is autoinhibited by tight binding to GBD (GTPase-binding domain). However, the attachment of Cdc42 to GBD induces conformational changes leading to the exposure of the VCA and subsequently to activation of Arp2/3 [40]. WAVE, on the other hand, can usually be found in the form of an inactive complex of five different proteins. They are mostly responsible for processing the signals that regulate WAVE activity. One of such signals is binding to Rac1 [41]. The depolymerization activity of CFL is also controlled by RhoGTPases. The downstream of Cdc42/Rac and Rho are p65PAK and p160ROCK. These proteins activate LIMK (LIM kinase), which by CFL phosphorylation reduces its filament decomposition activity. Research also indicates an association between Rac1/Pak1 (p21-associated kinase1) and LIMK1 (LIM kinase 1) signaling pathway in the control of CFL during lamellipodia formation [42]. Moreover, increased levels of phosphorylated Pak1, LIMK1, and CFL were found in the tissue samples from patients suffering from nonsmall cell lung cancer in comparison to those obtained from healthy individuals [43]. In turn, phosphatases from the Slingshot family dephosphorylates and thus reactivates CFL activity [44]. Similarly, some of the proteins from the formin family are regulated by Rho GTPases. These formins are classified as DRFs (Diaphanous-related formins), and in the resting state, they occur in autoinhibited form. This autoinhibition is caused by the intramolecular interaction between C-terminal and N-terminal domains and can be relieved as a consequence of the Rho GTPases action [45]. 

The role of actin and myosin in muscle contraction is well known. However, also in nonmuscle cells, the actomyosin complex is very important. The myosin superfamily is very large, however, most scientific reports focus on myosin II and III [46]. The increase in the length of filopodia depends on myosin IIIB [47], while their fusion requires the centripetal force produced by myosin IIA [48]. In addition to the participation of myosin in adhesion, migration, and maintaining the cell shape, myosin II, due to the formation of the cleavage furrow together with actin, is also involved in cytokinesis [49]. Myosins are not only cytoplasmic proteins but also nuclear ones. The presence of myosin I, II, V, VI, X, XVI, and XVIII was confirmed in the cell nucleus. Together with nuclear actin, they may be involved in transcription [50].

Thanks to ABPs and continuous polymerization and depolymerization, microfilaments create a specific architecture that allows them to perform many functions in the cell. Literature reports indicate the actin cytoskeleton function in different types of cell death, e.g., in apoptosis [51], autophagy [52], and entosis [53]. Unfortunately, in addition to its functions involving physiological processes, actin contributes to carcinogenesis, EMT (epithelial–mesenchymal transition), metastasis and, tumor vascularization [54,55]. Altered levels of ABPs such as α-actinin, villin, filamin, formin, CFL1, Arp2/3, GLS, TAGLN, or fascin were found in many types of cancers, which correlated with poor clinical outcome [25,29,43,56,57,58,59,60,61].

## 3. Actin and ABPs in Carcinogenesis

Carcinogenesis is the multistep process, which includes initiation, promotion, and progression. During initiation, the DNA of a normal cell mutates under the influence of carcinogens (chemical, biological, or physical DNA-damaging factors) or spontaneously. As a result, oncogenes may undergo constant activation. It leads to an increase in the level of factors responsible for cell proliferation and inhibition of apoptosis (promotion). During the next step, the tumor gains malignant properties. Enhanced angiogenesis, invasion, and metastasis facilitate tumor progression [62]. Literature reports indicate that alterations at the actin level may also be important in the process of DNA repair, chromatin remodeling, or activation of oncogenes, necessary in the initial stages of carcinogenesis [63,64,65]. In this context, nuclear ABPs, i.e., nesprin-1 or CFL seem to be particularly interesting as their level may affect the expression of some genes [65].

### Actin in the Nucleus

However, the presence of actin in the cell nucleus is no longer controversial, it seems that not all of the functions of the nuclear actin and ABPs are well understood. Earlier doubts were related to the size of the nuclear actin filaments and difficulties in their detection. Current scientific reports suggest that nuclear actin may exist in many forms ranging from monomers to short polymers. Moreover, nuclear actin filaments can also be visualized using phalloidin. Under stress conditions induced by etoposide and doxorubicin, a filamentous form of actin was observed in the nuclei of the human leukemia HL-60 and K-562 cell lines [66]. Additionally, quantum dots proved to be an effective technique enabling the localization of F-actin in the cell nucleus at the ultrastructural level [67].

Cell homeostasis depends on the appropriate proportions between nuclear and cytoplasmic actin. It requires efficient transport between these two compartments through the nuclear pores. However, actin itself does not contain NLS (nuclear localization signal). Thus, it first has to bind with transporter proteins. Attachment of ABPs such as CFL or profilin enables actin transport by IPO9 (importin-9) and XPO6 (exportin-6) in and out of the cell nucleus, respectively [27,68]. Nuclear actin is a component of chromatin-remodeling complexes, including INO80 (Inositol-requiring mutant 80), BAF (BRG1/BRM-associated factor), and SWR1 (SWI2/SNF2-Related 1 Chromatin Remodeling Complex), and thus may affect transcription, as well as DNA replication and repair [69,70]. Additionally, it is associated with all three RNA polymerases [71]. The regulation of nuclear actin also depends on ABPs. Yoo et al. demonstrated that N-WASP induces nuclear actin polymerization. Additionally, the Arp2/3 complex is also involved in the formation of actin nuclear polymers, and by binding with RNA polymerase II, it regulates transcription [72]. Apart from participation in the import of actin to the cell nucleus, CFL is also involved in the regulation of the dynamics of nuclear F-actin assembly, as well as in polymerase II transcription by controlling the polymerization of actin [73,74]. Important ABPs that regulate the nuclear functions of actin are also FMN2 (formin-2), SPIRE1/2 (protein spire homolog 1/2), and GLS family proteins [64,75].

Nesprins are a family of large multidomain proteins that link the nuclear envelope to the cytoskeleton and nucleoskeleton. This group of proteins is encoded by SYNE (spectrin repeat containing nuclear envelope) genes and interacts with actin, lamins, emerin, and chromatin. Among them, nesprin-1, -2, -3 and -4 are distinguished [76]. Numerous reports indicate that nesprin-1 is downregulated in oncological diseases, while mutations in SYNE1 have been identified in different types of human cancers [77,78,79,80]. In turn, induced overexpression of nesprin-1 allows reversing the malignant phenotype of human liver cancer Huh7 cell line [77]. A nuclear membrane protein that contains actin-binding sites is also nesprin-2. It participates in the nuclear transport of proteins such as BRCA1 or NF-κB. Lower levels of nesprin-2 mRNA are found in breast cancer tissues [81]. Moreover, the protein may occur in different cellular locations. In the ovarian cancer cell line SKOV-3, nesprin-2 was located in the nuclear envelope, whereas in the Caov-3 cell line, in the cytoplasm. Importantly, in Caov-3 cells, the reduction of nesprin-2 was accompanied by a reduction of nuclear BRCA1 [82]. 

In addition, for villin, not only its level is important but also a dynamic balance between the nuclear and cytoplasmic localization is significant. Cytoplasm-nucleus translocation of the protein is associated with, e.g., Slug regulation. Slug is one of the cancer-related transcription factors, which favors cancer progression and metastasis [83].In turn, ectopic villin expression regulated by CDX2 (Caudal Type Homeobox 2) may be of great importance in the early stages of intestinal metaplasia and gastric cardia tumors [56]. However, neither villin nor CDX2 is related to none of the clinical and pathological parameters in colorectal cancer [84]. By contrast, another study showed that for colorectal cancer, CDX2 loss was an unfavorable prognostic biomarker but only in stage IV [85].

WASP is one of the ABPs involved in DNA repair. It binds to DSBs (double-strand breaks) and facilitates actin polymerization [86]. WASP accumulation was observed in the UO2S cell line after DSBs induction. Moreover, its interaction with the protein characteristic for DSBs (γH2AX) was also confirmed [87]. WASP may also be involved in the regulation of RNA polymerase II activity [88]. Additionally, the correlation between WASP and the p53/p21 signaling pathway also plays an important role in carcinogenesis [89], whereas the relationship between N-WASP and oncogenic KRas was observed in pancreatic ductal adenocarcinoma [90].

Additionally, literature reports indicate the importance of other ABPs in the carcinogenesis. High levels of α-actinin-4 correlate with the accumulation of β-catenin in the nucleus and the upregulation of genes responsible for tumorigenesis in cervical cancer [91]. The conclusion was supported during the database analysis. Moreover, the researchers suggest the involvement of α-actinin-4in chemoresistance [58]. In turn, SATB1 (special AT-rich sequence-binding protein-1) is primarily responsible for the regulation of gene expression through changes in chromatin architecture. Moreover, the SATB1/F-actin complex is involved in active cell death in the MCF-7 cell line [92]. In the case of ARPC2 (a subunit of the Arp2/3 complex), a high level of the protein promoted breast cancer oncogenesis [93]. In turn, FlnA (filamin A) binds signaling molecules, whereas its nuclear localization acts as a tumor suppressor through regulation of the transcription factors [94]. FlnA in the nucleus inhibits ribosomal RNA transcription by interaction with RNA polymerase I [95]. In turn, in prostate cancer, nuclear FlnA interacts with androgen receptor and inhibits the transcription of its target gene [96]. Moreover, the level of FlnA is associated with the expression of BRAC1 [97]. An exemplary diagram of the carcinogenesis process with ABPs involvement is presented in Figure 2.

## 4. The Involvement of Actin and ABPs in Cancer Progression (EMT and Metastasis)

Metastasis is the leading cause of cancer-related deaths and represents an important step in the subsequent course of carcinogenesis (Figure 2). A key driver of this process is EMT, during which cells change their phenotype from epithelial to mesenchymal and acquire motor skills. The markers of the process are E-cadherin (epithelial phenotype) and N-cadherin (mesenchymal phenotype). However, during EMT, continuous polymerization and depolymerization of actin is also a pivotal element. The cortical actin fibers turn into actin stress fibers found in mesenchymal cells. Additionally, actin polymerization is associated with the formation of invasive protrusions like lamellipodia and filopodia [99]. Figure 3 shows the involvement of actin and ABPs in the invasion and migration of cancer cells.

Numerous scientific studies indicate the participation of GTPases from the Rho family, including Rho, Rac, and Cdc42 proteins, in the formation of the leading edge by inducing the accumulation of F-actin in the front of cells [39]. Argenzio et al. reported that by binding profilin-1, CLIC4(intracellular chloride 4 -channel) acts in the RhoA-mDia2-regulated signaling pathway. This interaction leads to enhanced cortical actin assembly and increased filopodia formation. Moreover, they point to the possibility of limiting formin-dependent filopodia by the agonist-induced CLIC4 translocation [100]. In turn, Shankar and Nabi indicated that depolymerization of the actin cytoskeleton with Cyt D (cytochalasin D) decreases the level of F-actin and upregulates E-cadherin in cancer cells. The observed increase in E-cadherin was associated with reduced Rho A activation. This led to the conclusion that actin remodeling may reverse EMT process and thus affects metastasis. [101]. Additionally, the reorganization of actin filaments and increased activity of Cdc42 and Rac during EMT were confirmed in oral squamous cell carcinoma [102]. Rac protein is strongly involved in the formation of lamellipodia and cell migration. On the other hand, Steffen et al. indicate that it is not required for spreading or filopodia formation. However, scientists point to its key role in the maintaining of cell polarity and migration [103]. 

Another ABP involved in the EMT process is Arp2/3. Immunohistochemical analysis of breast samples showed that ARPC2 expression was higher in cancer tissues compared to those from healthy individuals. Additionally, the level of expression correlated with the tumor stage and the occurrence of metastases to the lymph nodes. Moreover, ARPC2 silencing led to the inhibition of migration and colony formation and the induction of apoptosis in MDA-MB-231 cells. Furthermore, the ectopic expression of ARPC2 increased the level of mesenchymal markers—N-cadherin and vimentin. The results indicate that a high level of ARPC2 promotes EMT in breast cancer [93]. A similar positive correlation between the expression of Arp2/3 and malignancy was observed in glioblastoma samples [57]. Inhibition of the Arp2/3 complex in glioblastoma cells resulted in the loss of cell lamellipodia and polarity, which was associated with reduced migration and invasion [57]. Furthermore, Choi et al. presented the possibility of limiting the migration and invasion of cancer cells (lung, pancreas, and colon) by blocking the function of ARPC2 using pimozide [104].

The nuclear T-cell-specific transcription factor SATB1 (special AT-rich binding protein 1) is another EMT regulator. Qi et al. reported significantly higher levels of SATB1 in prostate cancer tissue samples and cell lines with high metastatic potential [105]. However, the overexpression of SATB1 was noticed in numerous malignancies, including lung, breast, ovarian, colorectal, and liver cancers [106,107,108,109,110]. Furthermore, the protein regulates the expression of E-cadherin and promotes cell invasion and migration. Thus, high SATB1 expression is generally associated with an aggressive phenotype, metastases, and poor prognosis in many cancer types. In turn, SATB1′s knockdown inhibited invasion and induced apoptosis of lung cancer cells [106]. Similarly, in colorectal cancer, SATB1′s depletion caused a change in the expression of EMT markers (E-cadherin and N-cadherin) [109]. 

Many scientific reports indicate that elevated CFL expression is also associated with tumor progression and metastasis. It was observed in, e.g., breast, gastric, prostate, colorectal cancers, and melanoma [111,112,113,114]. Moreover, its high level correlates with an increase in EMT markers. Hensley et al. showed that enhanced CFL level resulted in a statistically significant increase in the N-cadherin with a simultaneous reduction in E-cadherin expression in bladder cancer [115]. In turn, the silencing of CFL1 reorganized the structure of actin fibers and inhibited migration and invasion of gastric cancer in in vitro and in vivo studies. Moreover, in BGC-823 cells, the filopodia extended as CFL1 expression increased [114]. However, CFL is not only an element of the cytoplasm but may also occur in the cell nucleus. Bracalente et al. noted that the nuclear localization of CFL is associated with a worse outcome in melanoma patients [113]. The described studies indicate a multifunctional character of CFL in EMT. It promotes the creation of invasive structures such as filopodia. However, is also related to the regulation of gene expression as a result of the reorganization of actin in the nucleus.

Nuclear ABPs also include FlnA, which is a protein that cross-links nonmuscle actin fibers. Low nuclear FlnA levels are characteristic for colorectal adenocarcinoma and prostate cancer [58,116]. Additionally, the level of FlnA correlates with the clinical stage of cancer, lymph node metastasis, and poor prognosis [94]. In turn, cytoplasmic FlnA acts as a promoter in cancer invasion and metastasis. Zhang et al. presented that knockdown of FlnA resulted in a limitation of the proliferation, migration, and invasion of human melanoma cells. Similar results were obtained during in vivo studies where the tumors’ size was reduced compared to the control group [117]. It is in accordance with the observations of Ji et al. who confirmed that the downregulation of FlnA reduced the metastatic potential of breast cancer cells (MDA-MB-231) [118].

Actin filament cross-linker α-actinin is a protein involved in cell migration. The overexpression ofα-actinin-1 in mammary epithelial cancer cells promotes the movements by the reorganization of actin cytoskeleton and destabilization of E-cadherin-based cell–cell adhesion. Additionally, the high expression of α-actinin in breast cancer patients is associated with poor prognosis [119]. Compared to healthy controls, overexpression of α-actinin-4 was significantly associated with the degree of clinical advancement and the lymph nodes status [120]. Thus, serum α-actinin-4 level may be a clinical prognostic factor in patients with breast cancer. α-actinin-4 can also be a therapeutic target in the treatment of gastric cancer. Liu and Chu, after knockdown of the protein, noticed a reduction in cell migration and invasion as well as stabilization of cell–matrix adhesion, which reduced metastasis [121]. In cervical cancer cells, α-actinin-4 promoted cell proliferation by inducing EMT. This process was favored by the Snail upregulation, which was related to the Akt signaling pathway [91]. 

There are also a few reports indicating the involvement of other ABPs in the migration of cancer cells and EMT process. Tanaka et al. observed that GLS, can switch E- and N-cadherin conversion via Snail, while its downregulation leads to EMT in the MCF10A cell line [61]. In turn, nesprin-3 regulates lung cancer cell migration as its downregulation inhibited the movements of A549 cells [118]. Additionally, villin is associated with the acquisition of a mesenchymal phenotype in invasive cancer cells [122]. Moreover, the nuclear pool of villin regulates EMT in HCT-116, MDCK, Caco-2, and HT-29 cells by modulating the expression and activity of Slug [83]. In turn, the high level of TAGLN enhanced the migration potential of colorectal cancer cells, which correlated with poor prognosis [123]. Hao et al. obtained similar results for breast cancer patients [60]. Moreover, ABPs are also involved in creating invasive structures like filopodia, e.g., fascin. Han et al. described the fascin-specific small molecules, which inhibit binding between fascin and actin. It was associated with the limitation of migration and metastasis of cancer cells through the stabilization of F-actin structure [29]. 

## 5. Role of Actin in Angiogenesis and Vasculogenic Mimicry

One of the game-changers during the tumor development is the moment in which it acquires the ability to settle a completely new site. Data supported by Dillekås et al. indicate that metastasis is responsible for the majority of cancer-related deaths [125]. One of the characteristics of any cancer is uncontrolled cell division. The quickly multiplying cancer cells face the problem of supplying mass with oxygen and nutrients and draining off unnecessary metabolites. The solution is the vascularization of the tumor in the process of angiogenesis. Simultaneously, cancer cells that have a direct connection to the circulatory system can effectively move with the bloodstream and colonize new habitats. In response to hypoxia, but also mutations of tumor suppressor genes or activation of oncogenes, levels of angiogenic factors increase. VEGF (vascular endothelial growth factor) is the best-known substance with such properties. It stimulates endothelial cells to migrate and proliferate, which starts the process of angiogenesis. Other known factors produced by cancer cells which stimulate TECs (tumor endothelial cells) are bFGF (basic fibroblasts growth factor), TGFα and TGFβ (transforming growth factor α and β), TNFα (tumor necrosis factor), PDGF (platelet-derived endothelial growth factor), granulocyte colony-stimulating factor, placental growth factor, IL-8, hepatocyte growth factor, and EGF (epidermal growth factor). 

During the angiogenesis process, some severe alterations in the cell morphology occur. Those are changes in cell shape and loosening in the structure of intercellular connections. In addition, there are modifications in the actin structure from characteristic star-like pattern to highly polymerized stress fibers. Actin polymerization is also necessary for VEGF-mediated migration of ECs (endothelial cells). ECs express two VEGF tyrosine kinases receptors: VEGFR1/Flt-1 and VEGFR2/Kdr. Both of them are necessary for angiogenesis as their lack results in the death of murine embryos [126]. However, VEGFR2 probably plays a role in the early steps of angiogenesis, whereasVEGFR1 is involved in the later stage of vessel formation [126,127]. Moreover, the receptors may also occur in the form of the VEGFR1/2 heterodimer, whose activation correlates with enhanced cell migration and the formation of tubular structures on Matrigel but seems not to be associated with increased ECs proliferation [128]. Further, VEGF action involves the activation of MAPK (mitogen-activated protein kinases) family proteins such as not only ERK1/2 (extracellular signal-regulated kinase) but also SAPK1 (stress-activated protein kinase1) and SAPK2/p38 kinases. Although SAPK2/p38 inhibition resulted in blockage of the actin polymerization process and the formation of stress fibers, it was not observed in the case of ERK1/2 inhibition. Interestingly, SAPK2/p38 also participates in bFGF proangiogenic action. Another protein involved in VEGF-induced actin remodeling in endothelial cells is the SH2/SH3 domain-containing protein Nck. The use of a dominant negative inhibitor of Nck in endothelial cells leads to the inhibition of VEGF-induced migration-related phenomena, including the reorganization of the actin cytoskeleton [129]. Further studies indicated that Nck in endothelial cells exists in the form of a complex together with PAK (p21-activated kinase) and N-WASP. The VEGF application causes the phosphorylation of VEGFR2 tyrosine kinase, which, in turn, recruits the Nck/PAK/N-WASP complex. As a result, both PAK and N-WASP undergo activation. It intensifies PAK-mediated focal adhesion turnover and enhances actin polymerization through N-WASP action [130]. However, it should be highlighted that the activation of N-WASP in endothelial cells in response to VEGF may be associated not only with Nck but also with Rho GTPase Cdc42. N-WASP is a signal transmitter between Cdc42 and Arp2/3 complex [131]. Furthermore, loss of Cdc42 leads to reduced phosphorylation of its downstream effectors p21 protein Pak2 ((Cdc42/Rac)-activated kinase 2) and p21 protein Pak4 ((Cdc42/Rac)-activated kinase 4). Knockdown of either of these kinases is associated with a loss of pMLC, which, in turn, led to significant irregularities in the structure of the actin cytoskeleton [132]. The studies, however, concerned the embryonic development of blood vessel networks in normal tissues. Simultaneously, the involvement of Cdc42 in tumor angiogenesis remains elusive. However, the proven increase in the protein expression in endothelial cells in response to VEGF indicates its involvement in this process as well [133]. 

An additional related mechanism may be YAP/TAZ (yes-associated protein/PDZ-binding motif) signaling. In a study recently published by van der Stoel et al., it has been shown that the YAP/TAZ complex’s direct target is a protein strictly related to the focal adhesion—DLC1(deleted-in-Liver-Cancer 1) [134]. Although the authors focused rather on the effect of substrate stiffness, a two-hour treatment of HUVECs with VEGF led to YAP activation, which was sufficient to drive DLC1 expression. In turn, DLC1 or its absence is a factor influencing the organization of the actin cytoskeleton, which is also strictly connected with the focal adhesion and junctional pattern. As the authors indicated, DLC1 knockdown led to the production of prominent basal fibers of F-actin in HUVECs. Simultaneously, the cells kept their VE-cadherin-based cell–cell junctions. It is particularly interesting considering that YAP and TAZ activation through VEGF treatment is regulated by Rho GTPase signaling, e.g., by Rac1 [135]. One of the described mechanisms by which Rac1 impacts TAZ activation involves PAK. Although the impact of changes in the levels of individual elements of this chain on the structure of actin is still pending attention, there is more and more evidence for the involvement of YAP/TAZ in angiogenesis associated not only with the embryonic development of the blood vessel network but also tumor vascularization through a close relationship with the regulation of actin cytoskeleton remodeling [135,136].

TCMs show some characteristic features that distinguish them from normal ECs. Moreover, targeting TCMs seems to be particularly attractive therapeutic option due to their general genetic uniformity. As a consequence, the application of the drugs inhibiting angiogenesis may bring similar results for many types of cancer. What’s more, the high genetic stability of TECs indicates that they would probably not show significant drug resistance. It is not a new approach. In 2004, bevacizumab, a drug targeting VEGF signaling, was introduced. However, some studies showed that it may be associated with severe side effects like lethal hemoptysis or intestinal perforations [137]. Simultaneously, the application of the drug for some types of cancer brings only poor results. Considering the above, a search for new, safer ways to target TECs seems to be reasonable. Since actin reorganization is strictly involved in the angiogenesis process, it may become a new promising target in oncological patients. Studies conducted by del Valle-Pérez et al. showed that VEGF-induced angiogenesis in HUVEC cells may be inhibited by knockdown of FlnA, which is one of the ABPs responsible for actin reorganization. An additional factor associated with angiogenesis the pro-inflammatory environment [138]. Gagat et al. observed that tropomyosin overexpression stabilizes the actin cytoskeleton under pro-inflammatory conditions, leading to the maintenance of the normal structure of intercellular connections. These reports indicate the possibility of modulating the behavior of vascular endothelial cells by manipulating ABPs [139,140].

However, the creation of tubular structures capable of transporting blood along with nutrients and oxygen is not limited to angiogenesis. In recent years, more and more importance has been attributed to the phenomenon called vasculogenic mimicry (VM). Although the prevalence and importance of VM in cancer still divides the scientific community [141,142], evidence confirms that this process is associated with poor prognosis in cancer patients [143,144,145]. However, there is no detailed description of changes in the actin structure in the course of this phenomenon so far. However, there are some indications that it is also an element of VM. Studies showed that some of the VM-blocking substances base on the disruption of actin cytoskeletal integrity. Salinomycin suppresses the formation of tubular structures in trastuzumab-resistant HER2-positive breast cancer cell lines by destabilizing the actin cytoskeleton [146]. Zoledronic acid inhibits VM in LM8 osteosarcoma cells through disruption of the F-actin structure [147]. However, the manipulation of proteins associated with tumor progression have a similar effect. Maes et al. showed that knockdown of BNIP3 (B-cell lymphoma 2 (BCL-2) 19 kDa interacting protein 3) results in a complete blockage of tubular-like network formation on Matrigel in murine melanoma cells (B16-F10). Additionally, in this case, the loss of BNIP3 attenuated aggressive behavior through actin cytoskeleton remodeling [148]. This is not surprising considering that BNIP3 correlates with the level of integrin-associated protein CD47, whose downstream effectors are Rac1 and Cdc42.Moreover, studies on highly aggressive MDA-MB-231 breast cancer cell line revealed that after TAGLN silencing, the cells were less prone to create capillary-like structures. Although in this study the induced effect was related to an increased interleukin-8 uptake rather than a direct effect on the cytoskeleton, given the involvement of this protein in the progression of many types of cancer, it may be a suitable therapeutic target in cancer research [149]. Considering the changes in the actin structure observed on the leading edge of migrating cells, manipulation of Arp2/3 may also be effective. Although, so far, there are no reports on the effect of silencing of the protein on VM, due to its close involvement in the creation of invasive structures, it may become an attractive research target [14]. Additionally, the recent study by Skruber et al. points to the essential role of profilin-1 in actin filament assembly, especially on the leading edge [150]. These reports could provide the basis for further research on the link between ABPs manipulation and VM inhibition. Figure 4 shows the involvement of actin in angiogenesis and VM and ABPs as a potential therapeutic target.

## 6. Conclusions and Future Perspectives

In summary, metastasis contributes to the increase in the mortality of oncological diseases. In the context of the formation of secondary foci, conventional therapy affecting only the primary tumor site seems to be ineffective. Thus, the development of methods aimed at carcinogenesis, especially at cancer progression, metastasis, and vascularization may be crucial in the fight against cancer. Taking into account the involvement of microfilaments and ABPs in these processes, the actin cytoskeleton is an excellent target for therapy. Recent literature data indicate that an increase in the level of many ABPs involved in the reorganization of the actin cytoskeleton is associated with the induction of the EMT process, metastasis, and a worse prognosis for cancer patients. Hence, manipulation of ABPs expression may suppress cell proliferation, motility, and migration and sensitize cancer cells to drugs. Based on the literature, Arp2/3, CFL, SATB1, α-actinin, FlnA, and fascin are of particular interest. On the other hand, it seems attractive to explore the possibility of manipulating ABPs in the context of vascularization. For instance, disturbing the microenvironment of the secondary foci may also contribute to increasing the effectiveness of anticancer therapy. In this case, special attention should be paid to N-WASP, tropomyosin, and TAGLN. 

However, more research is required for the exploration of other ABP proteins as potential targets in crucial steps of carcinogenesis. Moreover, the actin cytoskeleton and its associated proteins are a difficult target in anticancer therapy. In contrast to the spindle-forming microtubules, where Taxol inhibits microtubule dynamics and thus reduces the growth of quickly multiplying cancer cells, the use of actin-focused drugs is not that simple. This is mainly due to the involvement of actin and ABPs, in many, sometimes mutually exclusive, processes, as well as the formation of contractile structures in heart and skeletal muscles. Numerous scientific reports indicate the possibility of limiting the proliferation and migration of cancer cells by manipulating the actin cytoskeleton. However, currently, these studies focus mainly on cell lines due to the frequent cardiotoxic effects of the therapies based on actin and ABPs regulation.

## Figures and Tables

**Figure 1 cells-09-02245-f001:**
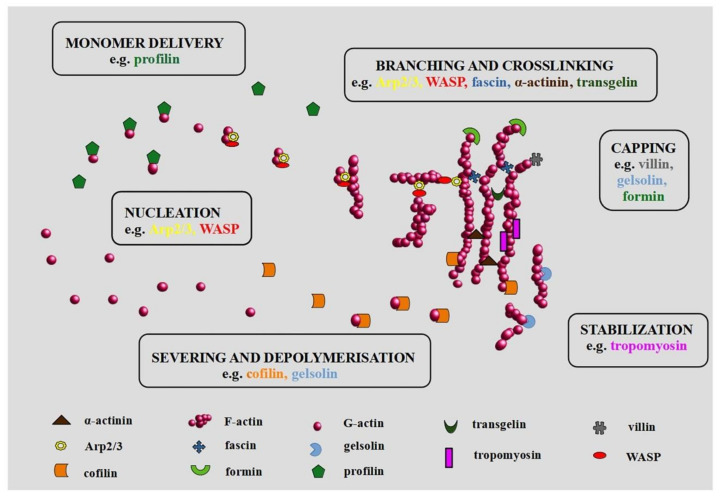
Actin-binding proteins (ABPs) in organization of actin network. Modified on the basis of dos Remedios et al. (2003) and Winder and Ayscough (2005) [4,5].

**Figure 2 cells-09-02245-f002:**
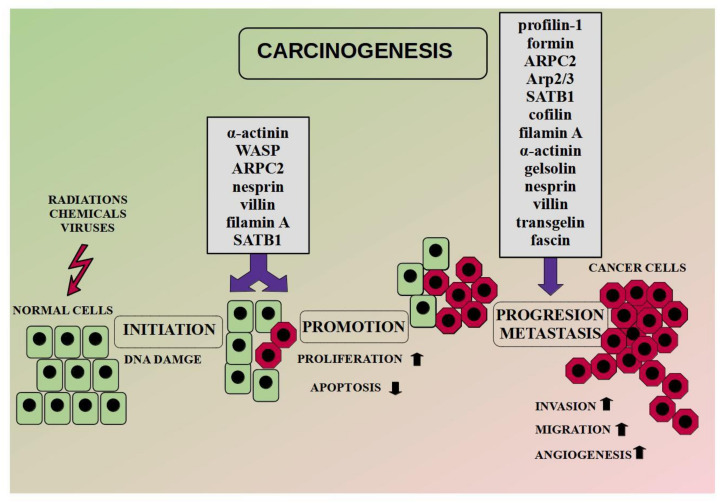
ABPs in carcinogenesis. Modified on the basis of Burgio and Migliore (2015) [98].

**Figure 3 cells-09-02245-f003:**
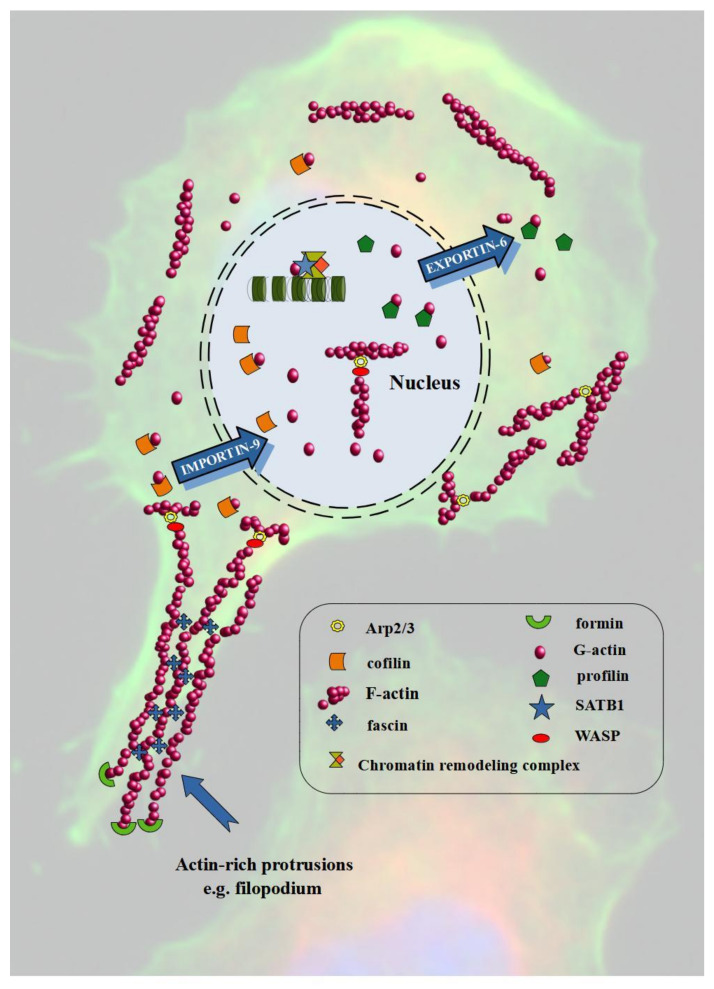
Actin and ABPs in cancer cells invasion and migration. Modified on the basis of Winder and Ayscough (2005) and Hurst et al. (2019) [5,124].

**Figure 4 cells-09-02245-f004:**
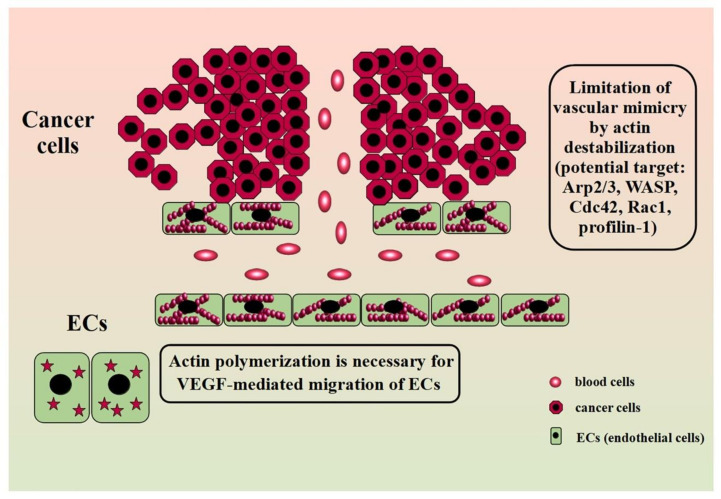
Actin in angiogenesis and vascular mimicry and ABPs as a potential therapeutic target. Modified on the basis of Lugano et al. (2020) [151].

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
