# Peer review of "Involvement of Actin and Actin-Binding Proteins in Carcinogenesis"

_cells, 2020, doi:10.3390/cells9102245_

Round 1
Reviewer 1 Report
Review
Izdebska et al. reviewed the literature about the role of actin binding proteins in malignant progression of cancer cells. This is a very interesting and often neglected topic and the authors clearly demonstrated that they are experts in this field. However, most researcher working on the field of carcinogenesis are not familiar with the regulation of actin dynamics.
Therefore, I suggest to re-structure the manuscripts as follows:
Chapter 2
Here, some principles of actin regulation are described. However, it is not complete.
Please describe in detail
- how Arp2/3 induces the formation of branched F-actin, including its regulation by NPFs class I (WASP etc.) and class II (cortactin). Role in formation of lamellipodia/invadopodia and migration/invasion
- describe how formins work (formation of linear F-actin and filopodia), including the proteins belonging to this family (mDia1-3, FLN etc.), its role in capping
- actin severing and capping proteins: gelsolin (controlled by calcium), cofilin (the 3 isoforms)
- actin bundling/cross linking proteins like Filamins and Fascin
- myosin (there are really a lot, also those involved in the regulation of the cleavage furrow)
- unique proteins, having diverse functions like Mena/VASP and villin
- thymosin beta 4 and profilin as G-actin binding proteins
- Role of the RhoGTPases cdc42, RhoA and Rac in stimulation of Arp2/3, formins and cofilin (here the involvement of PAK and slingshot)
After this chapter the authors should continue to mention ABDs whose increased expression correlates with bad clinical outcome, then focus on these proteins. In this context, Fascin is one the most important proteins.
Then: continue with chapter 3 and here include the sub-heading “actin in the nucleus”
Thereafter: Are there any clinical trials? For the fascin inhibitor G2 there is one
Finally: Discuss: why is the actin cytoskeleton hard to target? Compounds like taxol targeting microtubules are established chemotherapeutics why does this not work for actin. Because of the essential role of actin in heart and skeletal muscle
Minor issues:
Line 37: the cytoskeleton is not an organelle
Line 48: …stabilization of the actin ..
Line 72; F-actin is a long chain....
Line 128: ABPs seems to: Do the authors mean Nesprin-1?, Then, it should be mentioned here
General: Please explain all abbreviations as well as the role of these proteins (i.e SYNE1, ZBRK1, SATB1, CLIC4, SATB1). Introduce these proteins in chapter 2 as well as Nesprin 1 etc.
It would be good to include the work of Condeelis as well as Huang et al., 2015 (Fascin inhibitor)
Author Response
Dear Reviewer,
Thank you for all your valuable comments. All aspects of the opinion were carefully considered by our research team and the appropriate changes were introduced. The list of modifications is as follows:
Please describe in detail:
how Arp2/3 induces the formation of branched F-actin, including its regulation by NPFs class I (WASP etc.) and class II (cortactin). Role in formation of lamellipodia/invadopodia and migration/invasion
The rocess of branched F-actin structure formation with the participation of the Arp2/3 complex was described in Chapter 2 together with its regulation by WASP family proteins and cortactin. Moreover, its role in invasive structures formation was included. [Two of the seven subunits of the protein – Arp2 and Arp3 – are characterized by a great homology with G-actin...]
describe how formins work (formation of linear F-actin and filopodia), including the proteins belonging to this family (mDia1-3, FLN etc.), its role in capping
Formins were described in detail in Chapter 2. [Formins are also involved in actin polymerization...]
actin severing and capping proteins: gelsolin (controlled by calcium), cofilin (the 3 isoforms)
Gelsolin and cofilin descriptions were added together with its regulation or isoforms, respectively. [This is due to the sensitivity of segments 4-6 to calcium ions, whose binding causes a structural change in the protein...; Another aspect is the functional relationship of the ADF (actin-depolymerizing factor)/CFL family with environmental factors such as the presence of divalent ions or pH...]
actin bundling/cross linking proteins like Filamins and Fascin
Filamins and fascin descriptions were added in Chapter 2. [Proteins from the Fln family can bind both G- and F-actin...; In turn, fascin, through four specific tandem β-trefoil domains, attaches to actin filaments, taking part in their binding in parallel bundles...]
myosin (there are really a lot, also those involved in the regulation of the cleavage furrow)
The paragraph considering myosin interactions with actin and their role in cell migration process was introduced. [The role of actin and myosin in muscle contraction is well known...]
unique proteins, having diverse functions like Mena/VASP and villin
Mena/VASP and villin descriptions were added. [Interestingly, Wang et al. suggest the pro and anti-apoptotic impact of villin in small intestine epithelial cells... Another protein, Mena/VASP (mammalian enabled/vasodilator-stimulated phosphoprotein) is important during nucleation and polymerization of actin...]
thymosin beta 4 and profilin as G-actin binding proteins
Thymosin beta 4 and profilin were included as G-actin binding proteins. [The protein binds to G-actin, driving the filament assembly but only at the barbed end...; Another G-actin binding protein is Tβ4 (thymosin-β4), involved in the inhibition of F-actin polymerization.]
Role of the RhoGTPases cdc42, RhoA and Rac in stimulation of Arp2/3, formins and cofilin (here the involvement of PAK and slingshot)
Role of RhoGTPases in regulation of Arp2/3, fromins and cofilin was explained in detail. [An example could be the activation of the Arp2/3 complex by N-WASP and WAVE (WASP family Verprolin-homologous protein)...]
After this chapter the authors should continue to mention ABDs whose increased expression correlates with bad clinical outcome, then focus on these proteins. In this context, Fascin is one the most important proteins.
ABPs whose altered levels correlate with bad clinical outcome were listed in the end of Chapter 2 and further described in Chapter 3. [Altered levels of ABPs such as α-actinin, villin, filamin, formin, CFL1, Arp2/3, GLS, TAGLN, or fascin were found in many types of cancers which correlated with poor clinical outcome...]
Then: continue with chapter 3 and here include the sub-heading “actin in the nucleus”
According to the suggestion sub-heading describing actin in the nucleus was introduced in Chapter 3. [However, the presence of actin in the cell nucleus is no longer controversial, it seems that not all of the functions of the nuclear actin and ABPs are well understood.]
Thereafter: Are there any clinical trials? For the fascin inhibitor G2 there is one
Unfortunately, according to our knowledge there is no other clinical trials with ABPs inhibitors. However, information about fascin inhibitor clinical trial was added. [Due to the optimistic results of preclinical studies, NP-G2-044 fascin inhibitor is currently the only compound influencing the activity of ABPs that has entered the stage of clinical trials (NCT03199586)].
Finally: Discuss: why is the actin cytoskeleton hard to target? Compounds like taxol targeting microtubules are established chemotherapeutics why does this not work for actin. Because of the essential role of actin in heart and skeletal muscle
This issue was discussed in the conclusion section. [Moreover, the actin cytoskeleton and its associated proteins are a difficult target in anti-cancer therapy...]
Minor issues:
Line 37: the cytoskeleton is not an organelle
Line 48: …stabilization of the actin ..
Line 72; F-actin is a long chain....
Line 128: ABPs seems to: Do the authors mean Nesprin-1?, Then, it should be mentioned here
All of the corrections were included.
General: Please explain all abbreviations as well as the role of these proteins (i.e SYNE1, ZBRK1, SATB1, CLIC4, SATB1). Introduce these proteins in chapter 2 as well as Nesprin 1 etc.
All of the abbreviations were explained.
It would be good to include the work of Condeelis as well as Huang et al., 2015 (Fascin inhibitor)
Suggested paper was introduced to the manuscript.

Reviewer 2 Report
This review is a list of facts, not very well organized and presented, I mean, not very nice o read. I feel that many other reviews (for example in Nature Reviews) have done a better job on the same subject.
Comments:
- the only passage where the subject may be of interest for the community is lines 95-98 where the authors describe cytoplasmic actin versus nuclear actin. I therefore suggest that the authors propose a review on this subject and what is known, whether actin flows from the cytoplasm to the nucleus, whether other proteins do, etc In fact, I have often this questions from colleagues or at conferences: what about actin in the nucleus? How is it organized, where does it come from? Of course, the nucleus is small, and actin IN the nucleus is difficult to image, but a review on this subject, and on what are the remaining questions would be of better interest than the current version of this review.
- technical comments (the english and writing should have more attention):
- line 48: stabilization OF
- line 83: binds instaed of bounds
- CFL nor defined
- carcinogens not defined
- line 127 nuclear ABPs (remove "a"
- proteins should be written with a lower case
- end of line 165, remove "the"
- line 206: filament (no "s")
- Lines 212-226 are too long
- line 237 remove "s" in "cells"
- in general, instead of naming authors, name what is the result. Sometimes, the results are presented, sometimes eth authors are names. It would be easier if none of the authors are named, and if schemes would summerize known results (lots of proteins involved!!!).
- line 3030: a cancer cells (either remove "a" or "s"
- line 314 end of line, observed instead of observes
- Line 406 ansuitable (remove "n")
Author Response
Dear Reviewer,
Thank you for all your valuable comments. All aspects of the opinion were carefully considered by our research team.
We feel that participation of actin cytoskeleton and ABPs in the carcinogenesis process is very often neglected by specialists from the oncology field. Our work was aimed to summarize the latest and most important literature reports concerning the topic. We wanted to show how significant are actin and ABPs in every stage of cancer development and indicate some of the possible therapeutic targets for the future. We believe that this may bring broader attention to this topic and stimulate further studies.
However, we strongly agree that great interest arouses around the topic of nuclear actin. We will be happy to write a review paper on this topic in the nearest future. Moreover, in the current manuscript sub-heading describing actin in the nucleus was introduced.
The manuscript was corrected according to all of the technical comments and extensive language and grammar revision was done.
In general, we tried to focus more on the results and some of the authors' names were omitted where appropriate.

Reviewer 3 Report
Dear authors
The review entitled "Involvement of actin and actin-binding proteins in carcinogenesis" it is well done but needs a major grammar revision. Regarding to figures and bibliography it is my opinion that they are fine
Author Response
Dear Reviewer,
Thank you for all your valuable comments. According to your suggestion extensive language and grammar revision was done.
Round 2
Reviewer 1 Report
The manuscript has been improved and only few minor revisions are necessary:
Page 4, line 109. Please exchange “WASP-homology 2 domains” by “formin homology domains FH1 and FH2”
Page 11, line 329: there are two “empty brackets”
Page 12, line 337. Something is missing in this sentence
Reviewer 3 Report
Dear Authors
You have satisfied my requests